



# Reconstructing patterns of coastal risk in space and time along the US Atlantic Coast, 1970–2016

Scott B. Armstrong[1] and Eli D. Lazarus[1]

[1]Environmental Dynamics Lab, School of Geography & Environmental Science, University of Southampton, UK

*Correspondence to*: Eli Lazarus (E.D.Lazarus@soton.ac.uk); Scott Armstrong (S.B.Armstrong@soton.ac.uk)

**Abstract.** Despite interventions intended to reduce impacts of coastal hazards, the risk of damage along the US Atlantic Coast continues to rise. This reflects a long-standing paradox in disaster science: even as physical and social insights into disaster events improve, the economic costs of disasters keep growing. Risk can be expressed as a function of three components: hazard, exposure, and vulnerability. Risk may be driven up by coastal hazards intensifying with climate change,

or by increased exposure of people and infrastructure in hazard zones. But risk may also increase because of interactions, or feedbacks, between hazard, exposure, and vulnerability. Here, we present a data-driven model that describes trajectories of risk at the county scale along the US Atlantic Coast over the past five decades. We also investigate indications of feedbacks between risk components that help explain these trajectories. Our findings suggest that spatially explicit modelling efforts to predict future coastal risk need to address feedbacks between hazard, exposure, and vulnerability to capture emergent

patterns of risk in space and time.



## 1 Introduction

Risk reduction in developed coastal zones is a global challenge (Parris et al., 2012; Sallenger et al., 2012; Witze, 2018; Wong et al., 2014). In general terms, risk can be expressed as a function of hazard, exposure, and vulnerability (NRC, 2014; Samuels and Gouldby, 2009). Hazard is typically expressed as the likelihood that a natural hazard event will occur (e.g., a

recurrence interval for a storm of a given magnitude) or as a chronic rate of environmental forcing (e.g., a rate of sea-level rise). Exposure tends to capture either the economic value of property and infrastructure that a hazard could negatively impact, or the number of people a hazard could affect. Vulnerability can reflect a wide variety of dimensions, but in physical terms (relative to social metrics) vulnerability generally represents the susceptibility of exposed property to potential damage by a hazard event (NRC, 2014). Although the reduction of disaster risk – across all environments, not only coastal settings –

is an intergovernmental priority (UNISDR, 2015), a paradox has troubled disaster research for decades. Even as scientific insight into physical and societal dimensions of disaster events get clearer and more nuanced, the economic cost of disasters keeps rising (Blake et al., 2011; Mileti, 1999; Pielke Jr. et al., 2008; Union of Concerned Scientists, 2018).

There are a number of possible explanations for this trend. Economic costs could be rising because natural hazards,

exacerbated by climate change, are getting worse (Estrada et al., 2015; Sallenger et al., 2012); because with migration and population growth more people are living in hazard zones (NOAA, 2013); or because more infrastructure of economic value, from highways to houses, now exists in hazard zones (AIR Worldwide, 2013; Desilver, 2015; Union of Concerned Scientists, 2018). These drivers are typically addressed separately – but they are not mutually exclusive.

An alternative explanation for the disaster paradox is that environmental, population, and infrastructural drivers are systemically intertwined, resulting in "disasters by design" (Mileti, 1999) – unintended consequences of coupled interactions, or feedbacks, between natural forcing and societal shaping of the built environment. An example of one such feedback is when infrastructure development in hazard zones destroys natural features that would otherwise buffer hazard impacts (e.g., the loss of coastal wetlands that would have absorbed storm surge) (Barbier et al., 2011; Temmerman et al., 2013). An

example of another feedback is when hazard defences stimulate further infrastructure development behind them – a phenomenon called "safe development paradox" (Armstrong et al., 2016; Burby, 2006; Keeler et al., 2018; McNamara and Lazarus, 2018; Werner and McNamara, 2007). While both feedbacks can increase hazard impacts without any change in natural forcing, climate change accelerates them.

Investigations of coastal risk tend to focus on case studies of hazard, exposure, and/or vulnerability (Smallegan et al., 2016; Taylor et al., 2015), or on projections of future risk (e.g., Brown et al., 2016; Hinkel et al., 2010; Neumann et al., 2015). Few examine patterns of risk across large spatial scales ($\sim10^2$–$10^3$ km) or retrospectively over longer time scales ($>10^1$ yrs). Here, we develop a data-driven model to investigate how hazard, exposure, and vulnerability may describe trajectories of risk in space and time along the US Atlantic Coast, from Massachusetts to South Florida, at the county-level for the past 47 years

(Fig. 1). Our findings suggest that spatially explicit modelling efforts to predict future coastal risk need to address feedbacks between hazard, exposure, and vulnerability to capture emergent patterns of risk in space and time.



## 2 Methods

Using the components of risk broadly defined by the US National Research Council (NRC, 2014; Samuels and Gouldby, 2009), we represent coastal risk as a function of time ($t$) with the expression:

$$R(t) = H \, E \, V, \qquad\qquad (1)$$

where $R$ is coastal risk, $H$ is natural hazard, $E$ is exposure, and $V$ is vulnerability. We define hazard ($H$) in terms of chronic shoreline erosion (as opposed to the likelihood of a hazard event). We define exposure ($E$) in terms of the total property value of owner-occupied housing units in US Atlantic coastal counties. We address vulnerability ($V$) as a function of beach width, modulated by beach nourishment – the active placement of sand on a beach to counteract erosion – which functions as a buffer between hazard and exposure (Armstrong and Lazarus, 2019; Armstrong et al., 2016). For the purposes of this

analysis, we limit our consideration to physical infrastructure; we do not address socio-economic or demographic vulnerability (Cutter and Emrich, 2006; Cutter and Finch, 2008; Cutter et al., 2006, 2008).

### 2.1 Hazard

We calculated rates of shoreline change in two different ways to compare their respective effects on risk over time.

### 2.1.1 Shoreline-change rates from shoreline surveys

First, we calculated "end-point" rates of change from surveys of shoreline position published by the US Geological Survey (USGS) (Himmelstoss et al., 2010; Miller et al., 2005). An end-point rate is the cross-shore distance between two surveyed shoreline positions, divided by the time interval between the surveys. Using the Digital Shoreline Analysis System (DSAS) tool for Arc GIS (Thieler et al., 2008), we cast cross-shore transects every 1 km alongshore to intersect the surveyed shorelines, and at each transect calculated the end-point rate for three time periods (Armstrong and Lazarus, 2019):

"historical", from the first survey to 1960; "recent", from 1960 to the most recent survey; and "long-term", from the first survey to most recent (Fig. 2a, e, i; Fig. 3a). We calculated the median historical, recent, and long-term rates of shoreline change for each county alongshore.

We used 1960 to differentiate between historical and recent shoreline-change rates because during that decade, beach nourishment overtook shoreline hardening to become the predominant form of coastal protection in the United States (NRC,

1995, 2014). Cumulative, diffuse effects of nourishment are therefore embedded in recent and long-term rates of shoreline change (Hapke et al., 2013; Johnson et al., 2015). A historical rate calculated from shorelines surveyed prior to 1960 may better reflect environmental forcing in the effective absence of beach nourishment (Armstrong and Lazarus, 2019). These historical rates are not "natural" rates: human alterations to the US Atlantic Coast began long before 1960, with engineered protection, including seawalls, groyne fields, and limited beach-nourishment projects (Hapke et al., 2013). Here, we consider

them a pre-nourishment "background" rate of chronic forcing.

### 2.1.2 Shoreline-change rates from sea-level change rates

To test an independent measure of chronic shoreline-change hazard, we also derived rates of shoreline change (Fig. 4a, e) from recorded rates of sea-level change (Holgate et al., 2013; PSMSL, 2018) and a USGS dataset of cross-shore slope for the US Atlantic Coast (Doran et al., 2017). We calculated spatially distributed rates of sea-level rise from annual tide-gauge





records maintained by the Permanent Service for Mean Sea Level (PSMSL) (Holgate et al., 2013; PSMSL, 2018). For each tide-gauge record, we linearly interpolated across gaps in the annual data. We smoothed the resulting continuous record with a 10-year moving average, and calculated the annual rate of sea-level change (Table S1). Because the tide-gauge locations are not evenly distributed alongshore, to find rates of sea-level change for the full extent of the US Atlantic Coast we linearly

interpolated rates of sea-level change between tide-gauge stations, and calculated the median annual rate of sea-level change at each coastal county. To convert a vertical change in sea level to a horizontal change in shoreline position, we shifted shoreline position at each transect up (or down) cross-shore slope from USGS coastal lidar surveys (Doran et al., 2017) (Table S2). Linking the slope measurements to county shapefiles with a spatial join, we calculated median slope per county and then the horizontal distance that each annual vertical change in sea level moved the shoreline (Fig. 4a).

The relationship between sea-level change and shoreline position is more complicated than the one abstracted in our deliberate simplification (Cooper and Pilkey, 2004; Lentz et al., 2016; Nicholls and Cazenave, 2010). Our estimation is effectively a "bathtub model" of change, controlled only by topography with no incorporation of wave-driven sediment transport or other shoreline dynamics. However, for this exercise, our method is useful for its simplicity – especially given the spatial scales under consideration – and for the independent estimation of shoreline change that it provides.

**2.1.3 Sign convention**

By the sign convention in our calculations, a negative rate of shoreline change denotes accretion (reducing hazard), and a positive rate denotes erosion (increasing hazard) (Fig. 2a, e, i). Hazard magnitudes are normalized by the minimum and maximum rates to range between 0–1.

**2.2 Exposure**

To represent exposure along the US Atlantic Coast, we used county-level Census data for the total value (adjusted to 2018 $USD) of owner-occupied housing units for each decade from 1970 (Table S3) (Minnesota Population Center, 2011). Because property value data are sparse for the 2010 Census community survey (16 Atlantic coastal counties are missing), we instead used the 2009–2013 Census five-year survey. Several five-year Census surveys incorporate 2010, but we chose the 2009–2013 survey because it provides full overage of all the Atlantic coastal counties, and its mean of total values is closest

to the 2010 Census community survey (for those Atlantic coastal counties surveyed in 2010). We adjusted the county-total values of owner-occupied housing units to 2018 $USD and divided by the number of transects in each county to yield a proxy for property value per alongshore kilometre. Because of the range of values along the coast, we took a log-transform and normalized the results to fall between 0–1 (Fig. 2 b, f, j; Fig. 3 b).

**2.3 Vulnerability**

We represented vulnerability ($V$) with a two-part relationship that tracks beach width ($V_{bw}$) and beach nourishment ($V_{bn}$) over time:

$$V = 0.5V_{bn} + 0.5V_{bw} \qquad\qquad (2)$$





Because the value of exposed property is not included in $V_{bw}$ or $V_{bn}$, this formulation disentangles vulnerability from exposure – a subtle but important conceptual departure from the definition used by the National Research Council (NRC, 2014; Samuels and Gouldby, 2009), which includes property values in vulnerability.

We made the beach-width component ($V_{bw}$) inversely related to vulnerability, such that vulnerability increases as beach
width decreases. We express the beach-width component as:

$$V_{bw} = (x_0 + 1) - x, \qquad (3)$$

where $x_0$ is maximum beach width and $x$ is beach width. We then normalized by the maximum and minimum $V_{bw}$. Because the real measurements are unavailable, we assumed that in 1970 all counties had the same beach width ($x$). From this baseline, the county-scale shoreline erodes or accretes according to the linear rate determined by the hazard condition
(historical, recent, long-term, or sea-level derived). Because we used counties as the smallest spatial unit of comparison, our assumption implies that each county is fronted by beach. The physical geography of the real coastline is, of course, more spatially heterogeneous. Our analysis is too coarse to capture, for example, change at isolated pocket beaches in a predominantly rocky coastline, but counties with rocky coastlines will reflect very low or null rates of shoreline change.

For the beach-nourishment factor ($V_{bn}$), we collated beach-nourishment projects since 1970 by county from the beach-
nourishment database maintained by the Program for the Study of Developed Shorelines (PSDS, 2017). We took $V_{bn}$ as the running total number of nourishment projects per county over time (summed annually), and normalized $V_{bn}$ by the maximum total number of projects across counties as of 2016 (i.e., the county that nourished the most has $V_{bn} = 1$ in 2016). Each county starts with $V_{bn} = 0$ in 1970, and $V_{bn}$ increases incrementally with every nourishment project within the county boundary. We initiated $V_{bn}$ in 1970 to match the Census data for exposure ($E$). Because 80% of beach nourishment projects
on the US Atlantic Coast have occurred since 1970, we excluded a relatively small number of events. To test the sensitivity of our vulnerability and risk results to the 1970 start date, we examined the relative effects of (1) initiating $V_{bn}$ from the first nourishment project in our record (in 1930), and (2) excluding the $V_{bn}$ term altogether (Fig. S1). Although the risk patterns resulting from these sensitivity tests changed in detail, their general characteristics did not.

In our routine, until a county nourishes for the first time, beach width ($x$) changes according to the county median linear
erosion rate ($\gamma$):

$$x(t) = x_{t-1} + \gamma_t \qquad (4)$$

The linear erosion rate ($\gamma$) applied to each county is either the (pre-normalised) historical, recent, or long-term shoreline change rate, or the rate derived from sea-level change, depending on the hazard scenario. The sign convention for $\gamma$ is negative for erosion, and positive for accretion.

Once a county has nourished – as determined by the empirical dataset of nourishment projects (PSDS, 2017) – beach width becomes a function of a linear erosion rate ($\gamma$), as in Eq. (4), and a nonlinear erosion rate ($\theta$), which is applied to the



nourished fraction of the total beach width (μ) to capture cross-shore and alongshore diffusion of nourishment deposition across and along the shoreface (Dean and Dalrymple, 2001; Lazarus et al., 2011; Smith et al., 2009):

$$x(t) = (1 - \mu)x_0 + \mu e^{-\theta t}x_0 + \sum_1^t \gamma_t, \tag{5}$$

where $x_0$ is maximum beach width, $\theta$ is nonlinear erosion rate, μ is the fraction of the total beach width that the nonlinear
rate applies to, γ is linear erosion rate, and $t$ is the number of years since the last nourishment project. If a county nourishes at least once in a given year, its beach is restored to a maximum width in that year before it begins to erode. (Our minimum temporal increment was 1 year, and we assumed that nourishment always occurs at the end of a given year.) Maximum beach width ($x_0$), nonlinear erosion rate ($\theta$), and the fraction of beach width affected by the nonlinear rate (μ) are variables applied to the full spatial domain. Beach width (at the county scale) thus changes at a linear rate (γ), where a negative value
is erosion and a positive value is accretion, with an additional nonlinear erosion rate ($\theta$) over a fraction of the beach (μ) when nourishment occurs, until the beach is restored to maximum width by a subsequent nourishment project or reaches a specified minimum width (here, 10 m). The $V_{bn}$ term is ultimately normalised by the maximum and minimum beach width.

Because vulnerability is normalised, the minimum beach width that we specify (10 m) affects the length of time it takes to reach maximum $V_{bw}$, but does not affect the overall magnitude of $V$. A wider minimum threshold means that $V_{bw}$ reaches a
maximum faster, and vice versa. We used a minimum width of 10 m to avoid the numerical instabilities in $V_{bw}$ that arise with a minimum width equal to or less than 0 m. The minimum width threshold does not affect the cumulative beach-nourishment factor.

We test the effect of altering $x_0$, $\theta$, and μ on both vulnerability and risk, under historical hazard and linear erosion rates (Fig. S1; Table S4). Sensitivity testing shows that vulnerability over time is highest in the case of a narrow beach ($x_0$ = 25 m) with
a high nonlinear erosion rate ($\theta$ = 0.75) affecting a large fraction of the beach (μ = 0.75). Vulnerability over time is lowest in the opposite case ($x_0$ = 100 m, $\theta$ = 0.05, μ = 0.25) (Fig S1). In calculating our results, we used a case in the middle of these extremes ($x_0$ = 50 m, $\theta$ = 0.5, μ = 0.33), applying a value of μ similar to the value (μ = 0.35) used by Smith et al. (2009) and Lazarus et al. (2011).

Like a ratchet, the cumulative beach-nourishment factor ($V_{bn}$) increases each time a county nourishes. The beach-width
factor ($V_{bw}$) is comparatively more dynamic, reaching a minimum after a nourishment project (as the wide beach buffers property from hazard) but increasing as the nourished beach erodes and coastal properties become more susceptible to hazard.

## 3 Results

### 3.1 Risk trajectories



Our data-driven model generates a pattern of coastal risk that varies in space and time at county scale along the US Atlantic Coast (Fig. 1). From 1970, each county generates its own risk trajectory that represents the interaction of hazard, exposure, and vulnerability in that county (Fig. 1 a). For visualisation and analysis, we scaled each county by the number of 1 km transects they comprise (Fig. 1 a). The result is a matrix of 2386 km over 47 years, in which each of the 2386 (1 km) rows is
associated with a county. Alongshore mean values for the whole US Atlantic Coast are taken from the full matrix so that they reflect the relative alongshore scale of each county (Fig. 1 b).

We find that the collective trajectory of risk increases from 1970 to 2016 for all hazard scenarios – despite the occurrence of 998 beach-nourishment projects, ostensibly intended to reduce risk, during the same period (Figs., 2, 3). The influence of beach-nourishment projects on vulnerability means that county-scale risk varies over time even if hazard forcing remains
constant. Because hazard based on measured shoreline change (historical, recent, and long-term) is spatially variable but temporally static (Figs. 2, 3), changes in risk over time under this model condition are driven by either exposure or vulnerability.

The overall risk trajectory also increases with the spatio-temporally variable hazard condition derived from rates of sea-level rise (Fig. 4). The alongshore mean rate derived from sea-level rise shows close agreement with the mean "recent" shoreline-
change rate, suggesting that our simplified "bathtub" representation of hazard is reasonable on a multi-decadal time scale (Fig. 5).

Individually, not all counties register rising risk trajectories over time. To compare how individual counties contribute to mean risk, we ranked each county ranked by its risk index in 2016 (Table 1). We also examined in detail two examples of how individual counties responded to different hazards and beach-nourishment cycles (Fig. 6). Plymouth County,
Massachusetts, demonstrates how vulnerability may respond to linear erosion rates ($\gamma$) that vary from eroding (negative, under the "historical" condition), to static (under the "long-term" and sea-level derived conditions), to accreting (positive, under the "recent" condition) (Fig. 6 a-d). Ocean County, New Jersey, demonstrates how the cumulative beach-nourishment factor ($V_{bn}$) can drive up risk (Fig. 6 e-h). There, $V_{bn}$ causes the local maxima and minima in vulnerability to increase over time (Fig. 6 g), such that even when beaches are at full width, exposed property is still subject to vulnerability $V > 0$. Ocean
County highlights how the cumulative beach-nourishment factor functions as a ratchet, forcing vulnerability to only increase over time. Because not every county practices beach nourishment, it is possible for a county to have $V = 0$ if its shoreline is accreting (e.g., Camden and McIntosh Counties, Georgia). A county that never nourishes will have a $V_{bn} = 0$, and if a county nourishes only once or twice then their $V_{bn}$ will remain negligible (but not negative). However, mean vulnerability is greater – and therefore mean risk is greater – when $V_{bn}$ is left out ($V = V_{bw}$) (Fig. S1 c, d), because its inclusion makes vulnerability
less sensitive to changes in beach width. For example, a county that does not nourish could have a narrow beach but a low $V_{bn}$, and therefore a lower vulnerability score than if its vulnerability were only a function of beach width.

Alongshore mean risk in our model also increases because of a well-documented national trend in exposure (NOAA, 2013). Exposure in an individual county may increase or decrease from one decade to the next, but mean exposure along the full span of the coast increases over time (NOAA, 2013; Union of Concerned Scientists, 2018). The 51 coastal counties in this
analysis represent 1.6% of all US counties, but since 1970 have constituted 6.9–9.25% of the total value of all owner-occupied housing units in the country (Fig. S2). Thus, while our data-driven model includes simplifying assumptions, we suggest that the increasing risk trends in our findings represent a real phenomenon, since exposure has risen at the coast




decade on decade in real terms, and our cumulative beach-nourishment factor both dampens mean vulnerability and highlights the reality of long-term risk in counties that nourish continually.

## 3.2 Component relationships

Finally, we compared the statistical distributions of exposure in high- and low-hazard counties, and in high- and low-
intensity nourishing counties (as an aspect of vulnerability), to examine whether the three components of risk, as we represent them, reflect temporal interrelationships.

To explore potential relationships between exposure and hazard, we sorted the exposure time series (Fig. 2) into counties associated with "high hazard" (eroding shorelines) and "low hazard" (accreting shorelines) for historical and recent shoreline change (Figs. 7, 8). We find that exposure increases each decade in zones of high and low hazard, alike, for both historical
and recent shoreline change (Figs. 7, 8). Under "historical" shoreline-change hazard, exposure of property value is greatest in zones of high hazard (Fig. 7 a-h, Fig. 8 a). Conversely, exposure to high hazard is relatively low for "recent" shoreline-change rates (Fig. 7 i-p, Fig 8 d), in part because recent shoreline-change rates tend to be less erosional than their historical counterparts (Fig. 3 a). The difference between relative distributions of exposure in high and low hazard zones for historical shoreline-change rates increases in significance decade on decade, with a decreasing Kolmogorov-Smirnov $p$-value that
reflects the significance of their divergence (Fig. 8 c). There is no such temporal divergence of exposure in high and low hazard zones for recent shoreline-change rates (Fig. 8 f).

To explore, in parallel, potential relationships between exposure and vulnerability, we sorted the exposure time series into nourishing and non-nourishing counties, and then by the intensity of beach nourishment (high or low) according to whether counties fell above or below the 2016 median value of cumulative $V_{bn}$ (Figs. 9, 10). We find that although exposure increases
each decade in nourishing and non-nourishing counties, alike, more property is ultimately exposed in nourishing counties. Moreover, the mean value of that exposed property increases at a greater rate than in non-nourishing counties (Figs. 9 a-h, 10 a-c). Initially, all property is exposed in counties where nourishment intensity is present but low (their $V_{bn}$ sits below the 2016 median) – which we expect, because for counties to accrue enough nourishment events to match the 2016 median cumulative-nourishment factor requires time (Fig. 9 i, m). Exposure in intensively nourished counties (counties that accrue
enough nourishment projects to have $V_{bn}$ above the 2016 median) shows a marked increase in the 1980s (Fig. 10 d). Total exposure in intensively nourished counties overtakes total exposure in sparsely nourished counties by the 2010s (Fig. 10 e), such that more property ends up exposed in counties where nourishment intensity is high (Figs. 9i – p, 10 d-f).

Both of these temporal relationships in spatial patterns of exposure and hazard (Fig. 7) and exposure and vulnerability (Fig. 9) are likely two vantages of same feedback, catalysed by beach nourishment. Higher property value is exposed where
historical shoreline-change hazard was high (Fig. 7, a–d) and recent shoreline-change hazard is low (Fig. 7, m–p) because those places also practice relatively intensive use of beach nourishment (Fig. 11). The cumulative effect of beach nourishment may be sufficiently strong to mask "true" rates of shoreline change (Armstrong and Lazarus, 2019) – a defensive intervention that, by reducing apparent hazard, may spur further development (Fig. 9), increasing exposure and creating demand for additional protection (Armstrong et al., 2016).



## 4 Discussion and implications

Our data-driven, spatio-temporal model of risk along the US Atlantic Coast produces trajectories that vary in space and, on average, rise over time for all four chronic hazard scenarios that we test (Fig. 5). We know from the underlying data that real exposure increases over time, but we suggest that our modelled risk trajectories also reflect intrinsic feedbacks between
hazard, exposure, and vulnerability (Mileti, 1999). We find more property is exposed in counties with "high hazard" historical shoreline-change rates and "low hazard" recent shoreline-change rates (Figs. 7, 8), and that exposure has increased more in places that have practiced beach nourishment intensively (Fig. 9, 10). The spatio-temporal relationships that we show between exposure and hazard (Figs. 7, 8) and exposure and vulnerability (Figs. 9, 10) may reflect a feedback between coastal development and beach nourishment (Fig. 11) (Armstrong et al., 2016; Armstrong and Lazarus, 2019) – a
manifestation of the "safe development paradox" (Burby, 2006), in which hazard protections encourage further development in places prone to hazard impacts (Di Baldassarre et al., 2013; Lazarus et al., 2016; McNamara et al., 2015; Mileti, 1999; Smith et al., 2009; Werner and McNamara, 2007).

Our model is exploratory, and we reiterate its main caveats. Although there are many kinds of coastal hazard (e.g., storm impacts, flooding), we represented "chronic" hazard with shoreline-change rates that are spatially heterogeneous but
temporally static. An alternative derivation of shoreline change, from sea-level rise rates and simplified shore slopes, varies in both space and time, and yielded overall results similar to those returned by the "recent" shoreline-change scenario. Exposure in our model only accounts for the monetary value of owner-occupied properties in coastal counties, as captured by the US Census, thus excluding other potential measures of exposure (e.g., Cutter et al., 2006, 2008; Neumann et al., 2015; NRC, 2014; Samuels and Gouldby, 2009; Strauss et al., 2012) and requiring that we spatially aggregate our analysis to
county scales. Finally, our measure of vulnerability includes no method of shoreline protection other than beach nourishment, and its dynamics are underpinned by a set of broad assumptions: that beaches comprise shorelines at the county scale; that in 1970, all counties have the same initial beach width; that a beach-nourishment project always restores a beach to its full width; and that counties with intensive nourishment programmes may render themselves more vulnerable over time by masking a chronic erosion problem (Armstrong and Lazarus, 2019; Pilkey and Cooper, 2014; Woodruff et al., 2018). We
do not directly address alongshore spatial interactions within or between counties (Lazarus et al., 2011; Ells and Murray, 2012; Lazarus et al., 2016). Despite these assumptions, our model captures temporal interactions among the components of risk that ultimately yield large-scale spatial patterns similar to those identified in recent, fully empirical studies (Armstrong and Lazarus, 2019; Armstrong et al., 2016).

We suggest that models intended to test different coastal management policies, interventions, and scenarios should aim to
include feedbacks between hazard, exposure and vulnerability. In our data-driven model, traces of these feedbacks – and perhaps others – are likely embedded in the data we use. More detailed work at the intersection of theory and empiricism is necessary to resolve how feedbacks between hazard, exposure, and vulnerability dynamically affect each component of risk, and to explore how different management interventions may mitigate – or exacerbate – the "safe development paradox".





**Acknowledgements**

The authors thank Evan Goldstein, Julian Leyland, and James Dyke for helpful discussions. This work was supported by the NERC BLUEcoast programme (NE/N015665/2).

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



**Figures**

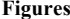

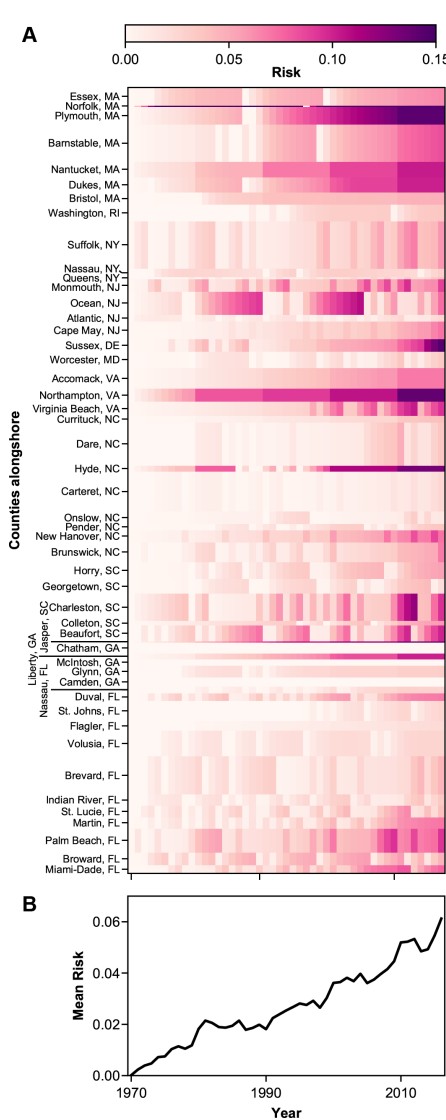

**Figure 1:** Evolution of risk (a function of hazard, exposure, and vulnerability) modelled at the **(a)** county scale along the US Atlantic Coast, from 1970–2016. Hazard in this simulation reflects historical erosion rates. County width is scaled by shoreline length. Note that risk in Norfolk County, MA, exceeds the maximum scale bar value of 0.15 (2016 risk = 0.418; see Table 1). **(b)** Mean risk through time, calculated from (a).





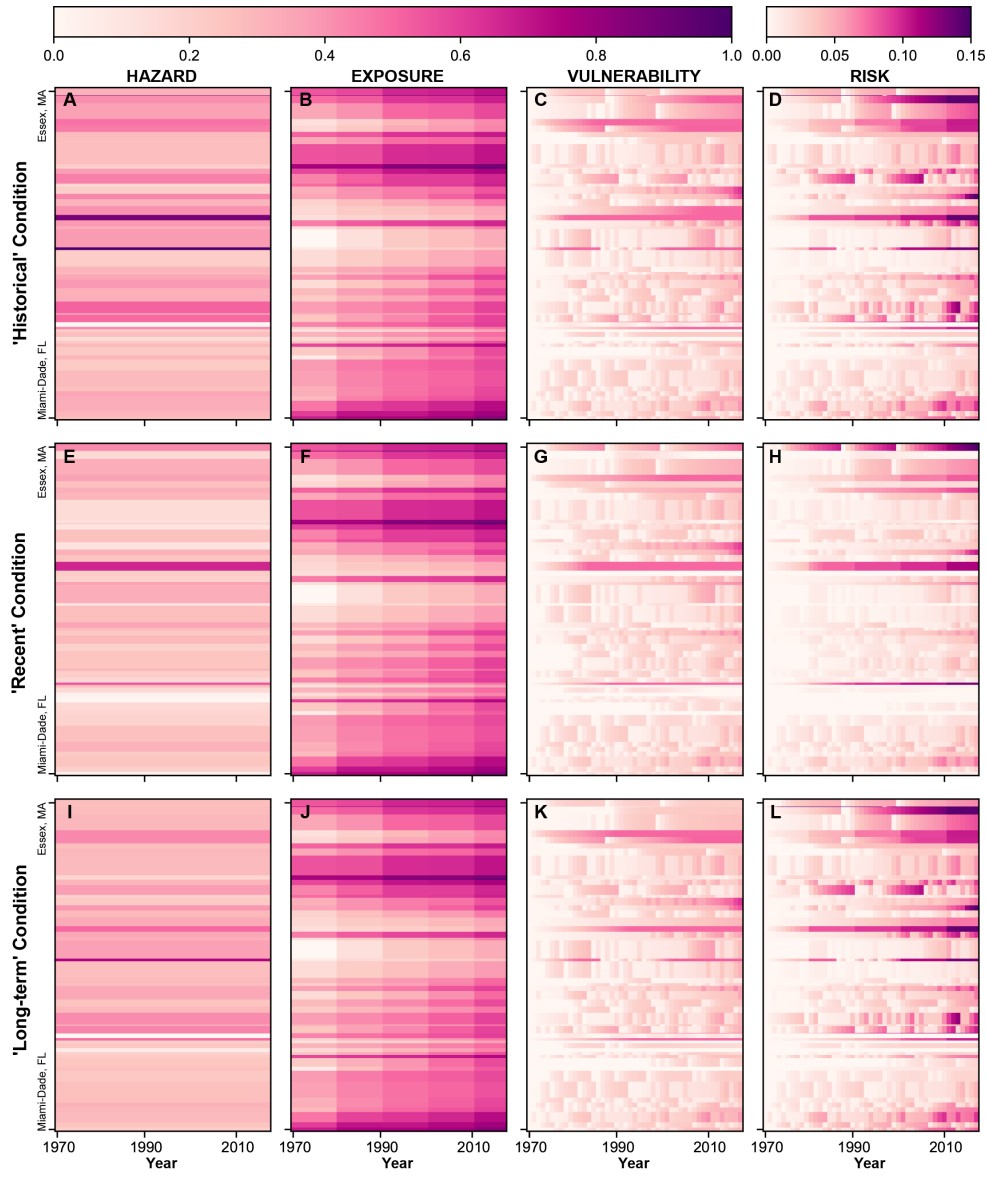

**Figure 2.** Columns show hazard, exposure, and vulnerability components and resulting risk. Each row of panels illustrates a different rate of shoreline change (i.e., hazard condition): **(a–d)** historical, **(e–h)** recent, and **(i–l)** long-term. Risk in Norfolk County, MA, exceeds the maximum scale bar value of 0.15 (2016 risk = 0.418; see Table 1)..





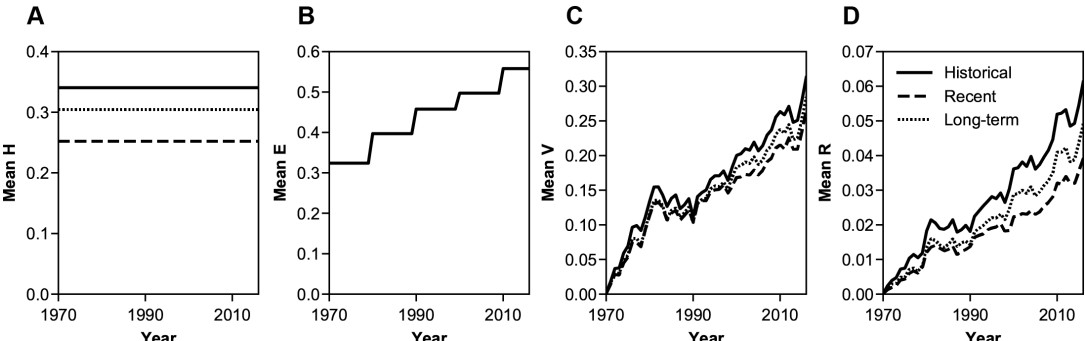

**Figure 3.** Evolution over time of alongshore mean risk components – **(a)** hazard, **(b)** exposure, and **(c)** vulnerability – and the resulting **(d)** mean risk, given historical (solid black), recent (dashed black), and long-term (dotted black) shoreline-change rates as hazard conditions.

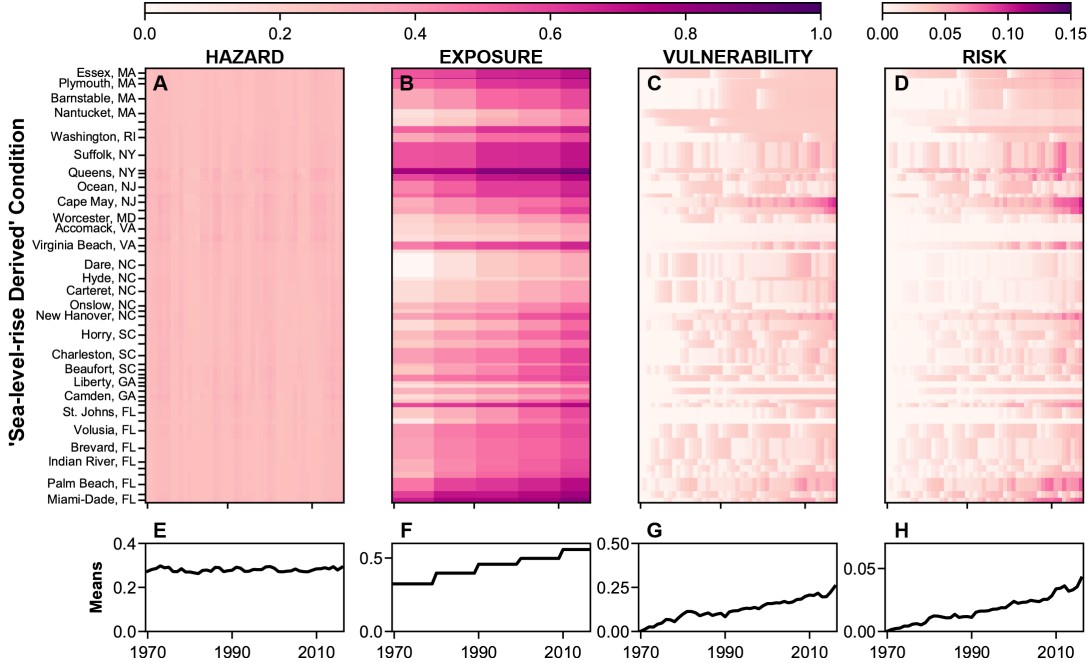

**Figure 4.** County-scale component **(a)** hazard, **(b)** exposure, **(c)** vulnerability and **(d)** overall risk evolution over time, and **(e–h)** corresponding means, using shoreline-change rates derived from sea-level change as the hazard condition.


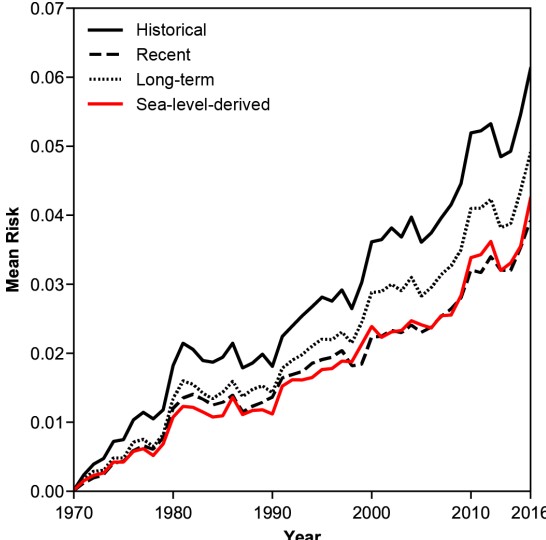

**Figure 5.** Comparative evolution of mean risk over time under different representations of shoreline-change rate (hazard condition): historical (solid black), recent (dashed black), long-term (dotted black), and sea-level-derived (red).

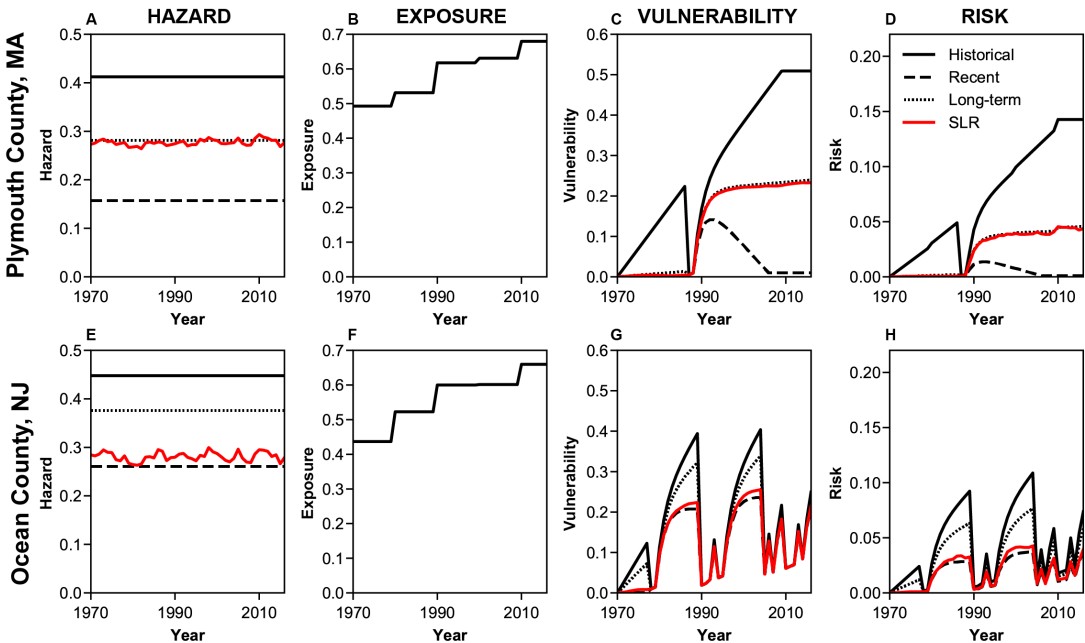

5   **Figure 6.** Evolution of **(a–c)** mean components and **(d)** risk for Plymouth County, Massachusetts, and **(e–h)** Ocean County, New Jersey. Line type indicates results under a given hazard condition. Note that the vulnerability time series for Ocean County (panel g) shows the "ratchet effect" of cumulative vulnerability from repeated beach nourishment episodes.





**Figure 7.** Distribution of exposed property, by decade, under **(a–h)** high and low historical and **(i–p)** high and low recent shoreline-change hazard. "High" hazard here is a value greater than 0.272 (the normalised value for a shoreline-change rate of zero); "low" hazard is a value greater than 0.272. High hazard therefore indicates erosion, and low hazard indicates accretion.





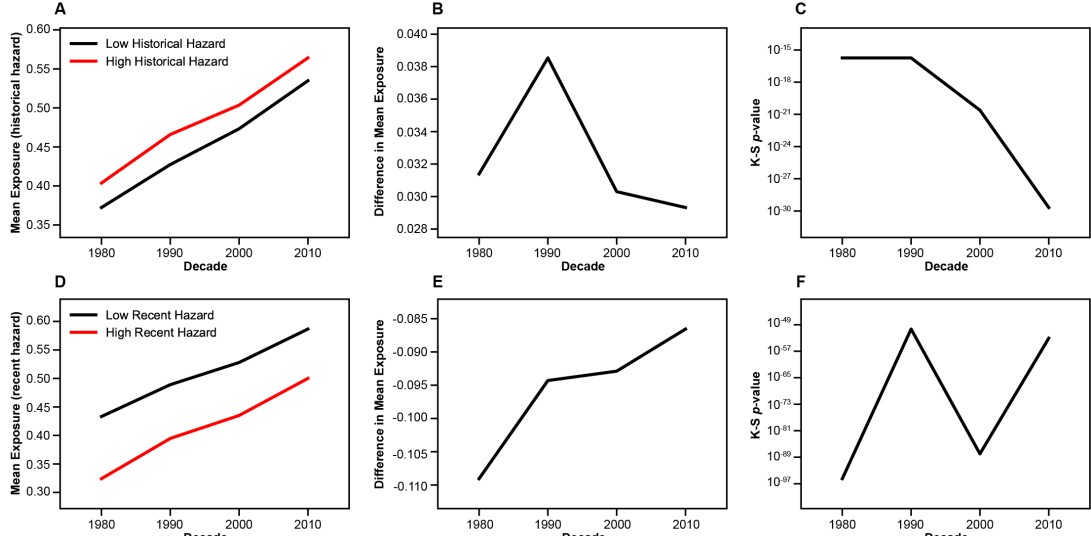

**Figure 8.** Comparisons of property exposed to high and low **(a–c)** historical and **(d–f)** recent shoreline-change hazard, from Figure 7. Columns show mean exposure each decade, the relative difference between mean exposure to high and low hazard each decade, and the Kolmogorov-Smirnov *p*-value for the difference in distributions each decade. All *p*-values indicate that the distributions are statistically distinct (i.e., a rejection of the null hypothesis that the distributions are sampled from the same parent distribution).





**Figure 9.** Distribution of exposed property, by decade, **(a–h)** in counties that have and have not nourished, and **(i–p)** in counties that have nourished above and below the 2016 median cumulative beach-nourishment index ($V_{bn} = 0.168$). The 2016 median $V_{bn}$ denotes the normalised value of the overall median cumulative number of nourishments across the domain.





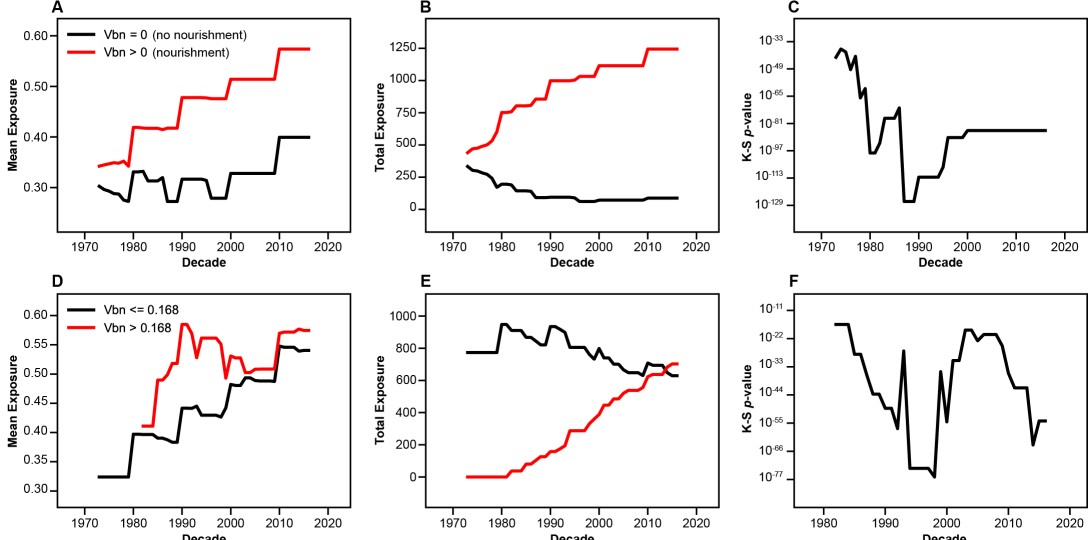

**Figure 10.** Comparisons of property exposed **(a–c)** in counties that have and have not nourished, and **(d–f)** counties that have nourished more or less than the 2016 median $V_{bn}$. Columns show mean exposure each decade, total exposure each decade, and the Kolmogorov-Smirnov $p$-value indicating the relative difference in exposure distributions each decade for each condition (nourished versus non-nourished; above versus below median $V_{bn}$). All $p$-values indicate that the distributions are statistically distinct (i.e., a rejection of the null hypothesis that the distributions are sampled from the same parent distribution).

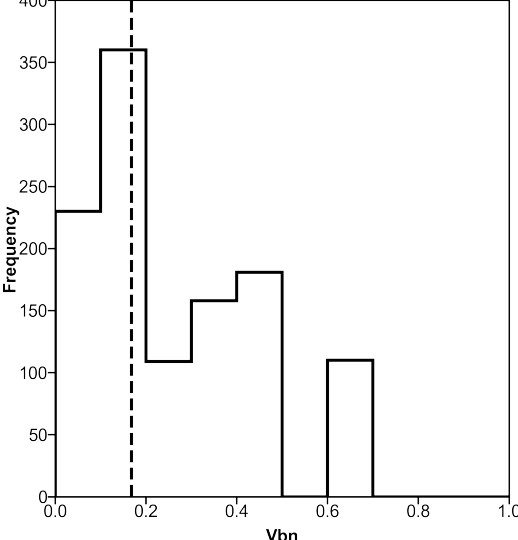

**Figure 11.** Cumulative beach-nourishment index ($V_{bn}$), as of 2016, at transects (across all counties) that express both high "historical" and low "recent" rates of shoreline erosion (see Fig. 7, a–d and m–p). Dotted line indicates the overall median $V_{bn} = 0.168$ in 2016 for the full domain. For this component distribution, median $V_{bn} = 0.178$ (mean = 0.251). This spatial correspondence between a major reversal in shoreline-change trend (from erosion to accretion) and above-average nourishment intensity is an indication of a coupling between chronic erosion (hazard) and defensive intervention (vulnerability).



## Tables

**Table 1. Counties ranked by risk in 2016, calculated with historic, long-term, recent, and sea-level-derived shoreline-change rates.**

| | Historical | | | Long-term | | | Recent | | | Sea-level-derived | | |
|---|---|---|---|---|---|---|---|---|---|---|---|---|
| Rank | County | State | 2016 Risk | County | State | 2016 Risk | County | State | 2016 Risk | County | State | 2016 Risk |
| 1 | Norfolk | MA | 0.4176 | Sussex | DE | 0.1303 | Essex | MA | 0.1451 | Cape May | NJ | 0.0995 |
| 2 | Sussex | DE | 0.1456 | Jasper | SC | 0.1176 | Liberty | GA | 0.1304 | Sussex | DE | 0.0899 |
| 3 | Plymouth | MA | 0.1427 | Liberty | GA | 0.1171 | Accomack | VA | 0.1130 | Miami-Dade | FL | 0.0809 |
| 4 | Northampton | VA | 0.1400 | Hyde | NC | 0.0999 | Sussex | DE | 0.1010 | Palm Beach | FL | 0.0807 |
| 5 | Jasper | SC | 0.1382 | Dukes | MA | 0.0946 | Bristol | MA | 0.0867 | Queens | NY | 0.0763 |
| 6 | Hyde | NC | 0.1328 | Nantucket | MA | 0.0924 | Nantucket | MA | 0.0790 | Duval | FL | 0.0661 |
| 7 | Nantucket | MA | 0.1026 | Beaufort | SC | 0.0828 | Palm Beach | FL | 0.0696 | Monmouth | NJ | 0.0647 |
| 8 | Liberty | GA | 0.1009 | Virginia Beach | VA | 0.0808 | Currituck | NC | 0.0682 | Virginia Beach | VA | 0.0640 |
| 9 | Dukes | MA | 0.1008 | Palm Beach | FL | 0.0806 | Queens | NY | 0.0642 | Norfolk | MA | 0.0637 |
| 10 | Beaufort | SC | 0.1002 | Northampton | VA | 0.0798 | Barnstable | MA | 0.0634 | New Hanover | NC | 0.0621 |
| 11 | Charleston | SC | 0.0953 | Cape May | NJ | 0.0787 | Brunswick | NC | 0.0497 | Suffolk | NY | 0.0613 |
| 12 | Virginia Beach | VA | 0.0949 | Charleston | SC | 0.0732 | New Hanover | NC | 0.0488 | Brunswick | NC | 0.0529 |
| 13 | Palm Beach | FL | 0.0940 | Monmouth | NJ | 0.0700 | Atlantic | NJ | 0.0435 | Martin | FL | 0.0512 |
| 14 | Monmouth | NJ | 0.0895 | New Hanover | NC | 0.0700 | Brevard | FL | 0.0420 | Beaufort | SC | 0.0495 |
| 15 | Barnstable | MA | 0.0841 | Suffolk | NY | 0.0618 | Washington | RI | 0.0419 | Charleston | SC | 0.0490 |
| 16 | Miami-Dade | FL | 0.0758 | Brunswick | NC | 0.0610 | Indian River | FL | 0.0412 | Atlantic | NJ | 0.0484 |
| 17 | Ocean | NJ | 0.0737 | Ocean | NJ | 0.0583 | Virginia Beach | VA | 0.0405 | Horry | SC | 0.0483 |
| 18 | New Hanover | NC | 0.0711 | Martin | FL | 0.0549 | Colleton | SC | 0.0403 | Nassau | FL | 0.0467 |
| 19 | Cape May | NJ | 0.0711 | Norfolk | MA | 0.0542 | Charleston | SC | 0.0389 | Essex | MA | 0.0463 |
| 20 | Martin | FL | 0.0708 | Queens | NY | 0.0514 | Cape May | NJ | 0.0366 | Nassau | NY | 0.0461 |
| 21 | Accomack | VA | 0.0694 | Miami-Dade | FL | 0.0497 | Ocean | NJ | 0.0365 | Brevard | FL | 0.0456 |
| 22 | Duval | FL | 0.0692 | Colleton | SC | 0.0481 | St. Lucie | FL | 0.0350 | Broward | FL | 0.0453 |
| 23 | Brunswick | NC | 0.0690 | Barnstable | MA | 0.0460 | Pender | NC | 0.0350 | Bristol | MA | 0.0444 |
| 24 | Essex | MA | 0.0639 | Plymouth | MA | 0.0457 | Martin | FL | 0.0330 | Volusia | FL | 0.0439 |
| 25 | Suffolk | NY | 0.0596 | Duval | FL | 0.0437 | Carteret | NC | 0.0328 | Plymouth | MA | 0.0438 |
| 26 | Colleton | SC | 0.0578 | Essex | MA | 0.0427 | Suffolk | NY | 0.0308 | Ocean | NJ | 0.0395 |
| 27 | Horry | SC | 0.0545 | Brevard | FL | 0.0419 | Dare | NC | 0.0302 | Washington | RI | 0.0382 |
| 28 | Bristol | MA | 0.0484 | Washington | RI | 0.0411 | Norfolk | MA | 0.0296 | Barnstable | MA | 0.0380 |
| 29 | Broward | FL | 0.0468 | Bristol | MA | 0.0397 | Beaufort | SC | 0.0287 | St. Johns | FL | 0.0376 |
| 30 | Brevard | FL | 0.0455 | Horry | SC | 0.0377 | Broward | FL | 0.0282 | Indian River | FL | 0.0372 |
| 31 | Queens | NY | 0.0415 | Broward | FL | 0.0377 | Worcester | MD | 0.0271 | Glynn | GA | 0.0371 |
| 32 | Currituck | NC | 0.0408 | St. Lucie | FL | 0.0354 | Horry | SC | 0.0252 | Carteret | NC | 0.0369 |
| 33 | St. Lucie | FL | 0.0402 | Indian River | FL | 0.0350 | Monmouth | NJ | 0.0225 | Pender | NC | 0.0360 |
| 34 | Pender | NC | 0.0370 | Dare | NC | 0.0348 | Dukes | MA | 0.0223 | Colleton | SC | 0.0321 |
| 35 | Washington | RI | 0.0364 | Accomack | VA | 0.0346 | Volusia | FL | 0.0190 | Chatham | GA | 0.0321 |
| 36 | Dare | NC | 0.0364 | Carteret | NC | 0.0333 | Nassau | NY | 0.0161 | St. Lucie | FL | 0.0318 |
| 37 | Worcester | MD | 0.0346 | Worcester | MD | 0.0323 | Onslow | NC | 0.0157 | Worcester | MD | 0.0312 |
| 38 | Indian River | FL | 0.0344 | Pender | NC | 0.0317 | St. Johns | FL | 0.0156 | Dukes | MA | 0.0275 |
| 39 | Nassau | NY | 0.0314 | Currituck | NC | 0.0315 | Georgetown | SC | 0.0155 | Nantucket | MA | 0.0274 |
| 40 | Glynn | GA | 0.0311 | Atlantic | NJ | 0.0303 | Chatham | GA | 0.0143 | Dare | NC | 0.0253 |
| 41 | Nassau | FL | 0.0276 | Volusia | FL | 0.0299 | Miami-Dade | FL | 0.0079 | Hyde | NC | 0.0190 |
| 42 | Volusia | FL | 0.0271 | St. Johns | FL | 0.0287 | McIntosh | GA | 0.0057 | Georgetown | SC | 0.0188 |
| 43 | Atlantic | NJ | 0.0268 | Nassau | NY | 0.0222 | Glynn | GA | 0.0011 | Onslow | NC | 0.0132 |
| 44 | St. Johns | FL | 0.0260 | Glynn | GA | 0.0184 | Plymouth | MA | 0.0010 | Camden | GA | 0.0083 |





| Rank | Historical County | State | 2016 Risk | Long-term County | State | 2016 Risk | Recent County | State | 2016 Risk | Sea-level-derived County | State | 2016 Risk |
|---|---|---|---|---|---|---|---|---|---|---|---|---|
| 45 | Carteret | NC | 0.0248 | Georgetown | SC | 0.0182 | Nassau | FL | 0.0008 | Northampton | VA | 0.0078 |
| 46 | Flagler | FL | 0.0223 | Nassau | FL | 0.0170 | Hyde | NC | 0.0006 | Jasper | SC | 0.0069 |
| 47 | Georgetown | SC | 0.0206 | Onslow | NC | 0.0128 | Flagler | FL | 0 | Liberty | GA | 0.0061 |
| 48 | Onslow | NC | 0.0136 | Chatham | GA | 0.0007 | Duval | FL | 0 | Accomack | VA | 0.0058 |
| 49 | Chatham | GA | 0.0005 | Flagler | FL | 0 | Camden | GA | 0 | McIntosh | GA | 0.0053 |
| 50 | Camden | GA | 0 | Camden | GA | 0 | Jasper | SC | 0 | Currituck | NC | 0.0050 |
| 51 | McIntosh | GA | 0 | McIntosh | GA | 0 | Northampton | VA | 0 | Flagler | FL | 0.0021 |