# Peer review of "Reconstructing patterns of coastal risk in space and time along the US Atlantic Coast, 1970–2016"

_Natural Hazards and Earth System Sciences, 2019_

## Referee Comment (RC1) · Anonymous Referee #1 · 1 Jul 2019

Manuscript Review: Reconstructing patterns of coastal risk in space and time along the US Atlantic Coast, 1970-2016 by S.B. Armstrong and E.D. Lazarus Submitted to Hydrology and Earth System Sciences, June 2019.

Paper summary: In this paper uses a model to investigate the interactions between exposure, vulnerability and hazard in aggregate coastal risk. The model uses data on shoreline changes, beach renourishment and housing value to estimate the effects of shoreline loss, sea level rise, and beach renourishment on coastal risk. Coastal risk is an important topic in a time of climate change when there are many, growing interactions between human and natural systems. Therefore, this paper addresses an

important subject in global change and coastal hazards.

General comments:

Overall, this paper is well written, but needs major work on the structure, data analysis and narrative. Major Issues:

1) Introduction speaks very broadly about risk and vulnerability, when the model is based on shoreline change and vulnerability that is defined by beach width and re-nourishment. Although this paper does contribute to the discussion about coastal risk, it is a very narrow framing of this problem (for example, your vulnerability measure does not include things like social vulnerability or built environment vulnerability). I think you need to reframe the introduction to talk about what this does do well – investigate geo-morphic risk and the impacts of renourishment - rather than what it doesn't.

2) Results need to be rewritten to talk about the major findings and not just describe the figures.

3) Data analysis is unclear at times (for example, how many counties are in the analy-sis? Do figures 7 and 9 show a count of counties or transects?). Please make sure the main details are discussed in the methods (like the statistical analysis in Figures 8 and 10).

4) To really hit home the interesting interactions between risk and renourishment, I think you need to add a spatial component – how does risk change in the model over the coastline, regionally (a color-coded map would be a great way to show this).

Specific Comments:

Title: I would reframe coastal risk throughout the paper as geomorphic or shoreline erosion risk.

Abstract: Overall: The abstract does not give any description of what kind of data goes into the model or major findings. If you reframed the abstract and intro around

geomorphic hazard and beach renourishment, this abstract could be a lot more specific and interesting. At the moment it is way too broad and says little about the actual study.

Line 12: What do you mean by "indications of feedbacks"?

Page 2: Introduction generally: It is good to start out broad with risk analysis, but the introduction does not really address geomorphic risk or beach renourishment. I think you could reframe in along these lines and set the reader up much better for what is coming. At the moment it stays too broad.

Lines 4-7: Add citations to these sentences

Lines 14-18: I think you can elaborate on this paragraph more and perhaps make it more about geomorphic change.

Line 24: Put the parenthetical and the citation in the same parentheses

Line 25: This paragraph needs more connective tissue and elaboration. Also needs to be focused more on geomorphic hazard. Could also talk about "levee effects" as another example of safe development paradox.

Lines 30-36: This paragraph still doesn't tell the reader what to expect going forward. Need to add objectives and specify that you are looking at shoreline change and beach renourishment as a subset of coastal risk.

Page 3: Line 7: How many counties?

Line 20: Are there different points in time for the first survey?

Page 4: Lines 20-28: The exposure you define here is capital exposure. Could you add in number of structures as a way to estimate count or some variable related to population number. Additionally, you are using the whole county as a way to get at the value of just beach front properties. This assumption could be wrong in places where there is less of a beach front community or tourism. You are assuming that the whole county is exposed to beachfront erosion.

Line 23: How are you dealing with uncertainty in this dataset? If there are whole counties missing, how much of other counties is missing? Would it be valuable to only use counties with low missingness?

Line 31: The vulnerability you are talking about here is just geomorphic vulnerability. These terms do not capture social vulnerability or built environment vulnerability. You should be clear about this from the beginning of the paper.

Page 5: Lines 7: What dataset is the beach width coming from?

Line 13: What about places that have a beach shoreline and then an interior shoreline (i.e. coastal lagoons)?

Line 18: Does the size of the renourishment play a role in the vulnerability change? Seems like the size would determine the change in vulnerability across the county.

Page 6: Result Overall: The results section needs to be rewritten. At the moment, much of it just describing figures without much narrative. Start your paragraphs out with a verbal description of the main finding for the paragraph and then get into the details. Additionally, there are results described here that are not discussed in the methods section (statistical analyses). Also some paragraphs are very short – two sentences in not a paragraph.

Page 7: Line 18: Table 1 is a great example of why this risk analysis needs to be reframed. Although Miami-Dade is subject to high probably of hurricane and king tides and high social vulnerability, it has a coastal risk of 0.08. This shows how this analysis is not a full reflection of total coastal risk, but more of geomorphic or shoreline erosion risks.

Lines 35-37: Your findings would be more realistic if you only included parts of the county that were most at risk of hazards from the coast and not the whole county.

Page 8: Line 11: Could you add some numbers about the peak of mode or the skewness to add some quantitative metrics to this description?

Lines 13-15: Add these statistical steps to the methods section.

Lines 25-27: Again add numbers about modes and skewness to add quantitative metrics to the description.

Lines 28-34: Much of this paragraph seems like discussion.

Page 9: Discussion overall: Again, this goes very broad and general for a study that is on geomorphic risk. I think you should really find those 2-3 results you want to highlight and talk about them here without taking it too broad. Would also consider making the analysis explicitly spatial (see comment below).

Lines 7-9: This is a great finding! And should be highlighted in the abstract.

Line 20: Also no measures of social or built environment vulnerability. . ..

Line 29: I think you could really hit this out of the park with some spatial discussion. What are the regional trends? What types of counties seems to be most at risk? A map color coded for risk would also be a great addition.

Figure 1: Why does it seem that some counties have more rows than others? Are you just labelling some of them? Or do some counties have multiple rows?

Figure 5: How are these rates calculated? What is the difference between historical and long-term?

Figures 7 and 9: How are the counts so high in these figures? If you only have 51 counties, shouldn't they be lower? Otherwise you are applying the same county level exposure number across multiple shoreline transects? I would suggest making these figures over counties and not transects. Additionally, is exposure normalized by area? Because otherwise this just shows that bigger counties with more resources for nourishment are adding more property, if I am reading this correctly (not sure if I am reading it correctly because of the count issue I brought up above).

Figure 8 and 10: The p-value part of the figures is a bit unnecessary. I think you could

use words in the results section to describe this. Also the y-axes, particularly in Figure 8, are misleading. It makes it look like the difference in exposure is huge, when in reality it is less than 0.1. Please make axes consistent (across historical and recent) and bigger range on the figure to better represent what is actually happening.

---

## Referee Comment (RC2) · Jorge Lorenzo-Trueba (Referee) · 18 Jul 2019

Amstrong and Lazarus discuss the evolution of risk in coastal communities along the East coast of the US. The authors defining risk as the product of hazard, exposure and vulnerability. In turn, they define each of these factors as a function of metrics that can be extracted from open source databases. This analysis, allows the authors to infer insightful information about the evolution of risk over the last few decades. Among other things, the authors find that there is significant correlation between the different factors that define risk, suggesting that feedbacks between hazard, exposure and vulnerability play an important role and should be taken into account. This manuscript conveys an

important message for the scientific community, as well as coastal managers and stake holders. I have a few comments aimed at improving the clarity:

Page 4-Line31: . . . that tracks the vulnerability associated with beach width (Vbw) and beach nourishment (Vbn). . . Page 5-Lines 5-10: Equation (3) suggests that Vbw is equal to 1 when x=xo. Is 1 just an arbitrary value? If this is the case, I suggest the authors clarify this in the text. Additionally, the authors normalize Vbw by the min and max of Vbw (as we can see in Figure 3 and 6, for instance). Being this the case, would it be easier to write the normalized expression as Vbw = 1-x/xo? Page 5-Line 7: . . . in 1970 all counties had the same beach width (x). . .. The use of "x" in this case might be misleading. I believe "x" is the beach width at any point in time, not just in 1970. Line 15-20: I suggest the authors include the equation used to calculate Vbn. Including this expression will also help to better understand lines 20-32 in the results section (page 7). Additionally, I suggest the authors better explaining why as beach nourishment volume and frequency increases, the vulnerability of a coastal community increases. I can see why this is the case, but it might not be intuitive. Is it perhaps due to the community becoming dependent on such practices, which in turn depend on the availability of a limited resource? Page 7 – Line 15: Would it be useful to mention here that the shoreline erosion rate predicted by bathtub models often underestimates the natural rate of erosion? This is particularly the case in barrier island environments, which are quite common in the region of study included in this manuscript. Page 7 – Line 18: . . .we ranked each county by its risk. . .

---

## Author Comment (AC1) · 16 Sep 2019

**RESPONSE TO REVIEWERS** – *nhess-2019-159*

13 September 2019

Dear Editors –

We thank you and the two reviewers for your time, and for the opportunity to revise this submission. Here we offer detailed responses to the comments made by both reviewers (whose remarks are *italicised*).

Sincerely –

*Eli Lazarus* (E.D.Lazarus@soton.ac.uk)
*Scott Armstrong*
* * *
**REVIEWER #1 (Anon) Comments**

*(1) Introduction speaks very broadly about risk and vulnerability, when the model is based on shoreline change and vulnerability that is defined by beach width and re-nourishment. Although this paper does contribute to the discussion about coastal risk, it is a very narrow framing of this problem (for example, your vulnerability measure does not include things like social vulnerability or built environment vulnerability). I think you need to reframe the introduction to talk about what this does do well – investigate geomorphic risk and the impacts of renourishment - rather than what it doesn't.*

We have sought to clarify our vantage and framing of the problem in the Abstract and Introduction (detailed replies below).

*2) Results need to be rewritten to talk about the major findings and not just describe the figures.*

We respectfully disagree with this characterisation of the Results section. However, in addressing R1's specific comments, we have also tried to improve the clarity of the text throughout, with an eye toward emphasising the major findings.

*3) Data analysis is unclear at times (for example, how many counties are in the analysis? Do figures 7 and 9 show a count of counties or transects?). Please make sure the main details are discussed in the methods (like the statistical analysis in Figures 8 and 10).*

We address these issues in detail where they appear, below.

*4) To really hit home the interesting interactions between risk and renourishment, I think you need to add a spatial component – how does risk change in the model over the coastline, regionally (a color-coded map would be a great way to show this).*

We suggest that this is what Figs. 1 & 2 show, albeit in a matrix rather than a spatially explicit map. (We revisit this point below.) The relationship between risk intensification in nourishment zones comes out most clearly in the shape of the statistical distributions (now Figs. 7 & 8), not in a "heat map" of their own – they are embedded in Figs. 1 & 2, but not easily seen in that format. Their collective statistical distributions are their signature.

*Title: I would reframe coastal risk throughout the paper as geomorphic or shoreline erosion risk.*

Noted – and see our reply, below, to the related comment on P2/Introduction.

We hew close to the formal definitions of hazard, exposure, and risk used by the US National Research Council. Those definitions are generic inasmuch as hazard, exposure, and vulnerability *types* can be substituted in. It is therefore unclear to us what "geomorphic risk" is. We can imagine geomorphic *hazard*, which is the role of shoreline change in this work. Or, because the definition of "vulnerability" is not always mutually exclusive from other components of risk, we can imagine geomorphic *vulnerability* where the susceptibility of exposed assets/people is a function of, for example, the recurrence interval of a geomorphic/natural event (e.g., hurricanes, floods, landslides). We treat vulnerability as a kind of buffer between hazard and exposure. But "geomorphic risk" is an entanglement of the three separate components (hazard, exposure, vulnerability), which we take pains to differentiate and not double-count in our analysis.

In that context, "coastal risk" is both generic and a readily accepted term. We agree with R1 that how coastal risk is then defined is essential. We have made amendments to the Abstract (see related comments below) to help make our framing/definition more readily accessible from the outset.

*Abstract: Overall: The abstract does not give any description of what kind of data goes into the model or major findings. If you reframed the abstract and intro around geomorphic hazard and beach renourishment, this abstract could be a lot more specific and interesting. At the moment it is way too broad and says little about the actual study.*

We have amended the Abstract (blue text) to read as follows:

"…But risk may also increase because of interactions, or feedbacks, between hazard, exposure, and vulnerability. Using empirical records of shoreline change, valuation of owner-occupied housing, and beach nourishment projects to represent hazard, exposure, and vulnerability, respectively, here we present a data-driven model that describes trajectories of risk at the county scale along the US Atlantic Coast over the past five decades. We also investigate quantitative relationships between risk components that help explain these trajectories. We find higher property exposure in counties where hazard from shoreline change has appeared to reverse from high historical rates of shoreline erosion to low rates in recent decades. Moreover, exposure has increased more in places that have practiced beach nourishment intensively. The spatio-temporal relationships that we show between exposure and hazard, and between exposure and vulnerability, indicate a feedback between coastal development and beach nourishment that exemplifies the "safe development paradox", in which hazard protections encourage further development in places prone to hazard impacts. Our findings suggest that spatially explicit modelling efforts to predict future coastal risk need to address feedbacks between hazard, exposure, and vulnerability to capture emergent patterns of risk in space and time."

*Line 12: What do you mean by "indications of feedbacks"?*

For clarity, corrected (P1, L14) to read "quantitative relationships" between risk components.

*Page 2: Introduction generally: It is good to start out broad with risk analysis, but the introduction does not really address geomorphic risk or beach renourishment. I think you could reframe in along these lines and set the reader up much better for what is coming. At the moment it stays too broad.*

We have addressed this with added text (blue) at the end of the Introduction (P2, L36):

"Here, we develop a data-driven model to investigate how hazard, exposure, and vulnerability may describe trajectories of risk in space and time along the US Atlantic Coast, from Massachusetts to South Florida, at the county-level for the past 47 years (Fig. 1). We restrict our analysis of risk to three specific components: shoreline change (hazard), valuation of owner-occupied housing units (exposure), and beach nourishment – the active, and typically repeated, placement of sand on a beach to counteract chronic erosion (vulnerability). We do not address socioeconomic or demographic exposure or vulnerability (Cutter and Emrich, 2006; Cutter and Finch, 2008; Cutter et al., 2006, 2008), nor the exposure of infrastructural aspects of the built environment beyond owner-occupied housing value. Neither do we address other types of coastal hazard, such as storm strikes or flooding, or types of hazard mitigation other than beach nourishment. Despite this tightly defined framing, our analysis captures underlying quantitative relationships between risk components. Our findings suggest that spatially explicit modelling efforts to predict future coastal risk need to address feedbacks between hazard, exposure, and vulnerability to capture emergent patterns of risk in space and time."

*Lines 4-7: Add citations to these sentences*

We have amended P2, L3, to emphasise that all of these definitions are those used by the US National Research Council.

*Lines 14-18: I think you can elaborate on this paragraph more and perhaps make it more about geomorphic change.*

We have fixed this issue by removing the paragraph break (P2, L19). Our discussion of geomorphic change is more appropriate later in the manuscript, as we move into specifics types of hazard.

*Line 24: Put the parenthetical and the citation in the same parentheses*

Amended for clarity.

*Line 25: This paragraph needs more connective tissue and elaboration. Also needs to be focused more on geomorphic hazard. Could also talk about "levee effects" as another example of safe development paradox.*

We have addressed this comment by amending the final paragraph of the Introduction (P2, L36) (excerpted above). We have also added references to the land-use-management and/or levee paradox (P2, L26): White, 1945; Burby & French, 1981; and Di Baldassare *et al.*, 2016.

*Lines 30-36: This paragraph still doesn't tell the reader what to expect going forward. Need to add objectives and specify that you are looking at shoreline change and beach renourishment as a subset of coastal risk.*

Changes to the final paragraph of the Introduction (P2, L36) likewise address this comment.

*Page 3: Line 7: How many counties?*

Amended (P3, L13) to state that we examine 51 coastal (ocean-facing) counties.

*Line 20: Are there different points in time for the first survey?*

We have amended this line (blue text), at P3, L29, to read:

"Because the dates of shoreline surveys vary by location, following Armstrong and Lazarus (2019) we calculate shoreline-change rates using the available surveys at each transect that are closest to the start- and end-date of each period. We calculated…"

*Page 4: Lines 20-28: The exposure you define here is capital exposure. Could you add in number of structures as a way to estimate count or some variable related to population number. Additionally, you are using the whole county as a way to get at the value of just beach front properties. This assumption could be wrong in places where there is less of a beach front community or tourism. You are assuming that the whole county is exposed to beachfront erosion.*

We agree that parcel-scale granularity of the housing stock would be optimal, especially because waterfront properties tend to be more expensive than inland ones. But using the county-scale data is not the same as assuming the whole county is directly exposed to erosion. It is simply an indicator of relative exposure. Because three factors contribute to risk, a hypothetical county like the one R1 is imagining – with high county-scale exposure but little beach-front development – may return a low risk index if its erosion rates are low and/or it doesn't nourish.

Still, in the context of beach nourishment (and hazard mitigation more broadly), oceanfront landowners are not the only people involved in mitigation actions. "Local governance" can include decision-making at the county scale, and governance processes vary by state. With the Census data available (which we use because they offer maximum spatial and temporal coverage of the domain), there is neither a perfect metric for exposure, nor an alternative metric that is unequivocally better than the one we choose (total value divided by length of county oceanfront).

*Line 23: How are you dealing with uncertainty in this dataset? If there are whole counties missing, how much of other counties is missing? Would it be valuable to only use counties with low missingness?*

We use the Census data with the most complete coverage available. The other areas of uncertainty in this analysis – the shoreline change measurements, our treatment of vulnerability – mean that whatever vagaries exist in the Census data are no more extreme than those inherent in the other components.

We do test the sensitivity of the analysis to those elements with the greatest potential to systematically affect our results. We present the various trajectories defined by different treatments of shoreline change rates, and by different parameters in our representation of temporal dynamics of beach nourishment. Through these tests, we understand what the model is doing – and that is perhaps the best way we can deal with uncertainty in this exploratory context.

*Line 31: The vulnerability you are talking about here is just geomorphic vulnerability. These terms do not capture social vulnerability or built environment vulnerability. You should be clear about this from the beginning of the paper.*

Now addressed by clarifications in the Introduction and Discussion.

*Page 5: Lines 7: What dataset is the beach width coming from?*

We do not use a dataset for beach width. At P5, L11, we state that "because real measurements are unavailable, we assumed that in 1970 all counties had the same beach width." There are no annual surveys of beach width for the US Atlantic Coast; this aspect of our model is parameterized based on best-available references.

*Line 13: What about places that have a beach shoreline and then an interior shoreline (i.e. coastal lagoons)?*

We now state at P5, L18, that we do not consider back-bay (interior) shorelines.

*Line 18: Does the size of the renourishment play a role in the vulnerability change? Seems like the size would determine the change in vulnerability across the county.*

We agree, but the nourishment dataset does not include volume for all projects over time (especially for projects farther back in time). We use total number of projects as a way to at least represent relative volume and/or renourishment activity over time. Indeed, not all nourishment projects involve the same volume of sediment, but the US does not (yet) use singular, Dutch-style "mega-nourishments" either.

*Page 6: Result Overall: The results section needs to be rewritten. At the moment, much of it just describing figures without much narrative. Start your paragraphs out with a verbal description of the main finding for the paragraph and then get into the details. Additionally, there are results described here that are not discussed in the methods section (statistical analyses). Also some paragraphs are very short – two sentences in not a paragraph.*

We appreciate the suggestion, but perhaps do not see the same issue. None of the topic sentences in our Results, as written, simply introduce figures. R1 notes, "Start your paragraphs out with a verbal description of the main finding for the paragraph and then get into the details," but by our reading, we do that. (Where there is perhaps more "narrative", R1 suggests in a comment below that the text be moved to the Discussion.)

We have added a statistical analysis section to the Methods, as recommended.

With our amendments to the framing of the argument and analysis, we have tried to more clearly motivate the Results overall.

*Page 7: Line 18: Table 1 is a great example of why this risk analysis needs to be reframed. Although Miami-Dade is subject to high probably of hurricane and king tides and high social vulnerability, it has a coastal risk of 0.08. This shows how this analysis is not a full reflection of total coastal risk, but more of geomorphic or shoreline erosion risks.*

We do not claim that this work is a "full reflection" of total coastal risk. We have aimed to better align readers' expectations with our changes to the Introduction.

*Lines 35-37: Your findings would be more realistic if you only included parts of the county that were most at risk of hazards from the coast and not the whole county.*

We agree – but see related comment above.

*Page 8: Line 11: Could you add some numbers about the peak of mode or the skew- ness to add some quantitative metrics to this description?*

We have added summary statistics in a supplementary table (Table S5.)

*Lines 13-15: Add these statistical steps to the methods section.*

Amended as suggested at P7, L8.

*Lines 25-27: Again add numbers about modes and skewness to add quantitative metrics to the description.*

As above.

*Lines 28-34: Much of this paragraph seems like discussion.*

Noted. As written, this paragraph ties off the findings and what they mean, setting up the wider implications in the Discussion. Given its technical detail, we have opted to leave it in the Results (P9, L18), rather than move it into the less technical Discussion.

*Page 9: Discussion overall: Again, this goes very broad and general for a study that is on geomorphic risk. I think you should really find those 2-3 results you want to highlight and talk about them here without taking it too broad. Would also consider making the analysis explicitly spatial (see comment below).*

We appreciate the suggestion – but again perhaps this comes down to a difference of stylistic preference. With the amended framing of the Introduction, the broader scope we consider in the Discussion should now be better anchored.

*Lines 7-9: This is a great finding! And should be highlighted in the abstract.*

We have amended the Abstract as suggested (excerpted above).

*Line 20: Also no measures of social or built environment vulnerability. . ..*

Amended (blue text) to read:

"…thus excluding other potential measures of exposure, such as socio-economic indices (e.g., Cutter et al., 2006, 2008; Neumann et al., 2015; NRC, 2014; Samuels and Gouldby, 2009; Strauss et al., 2012), and requiring that we spatially aggregate our analysis to county scales. Finally, our measure of vulnerability – intended to represent "susceptibility" (NRC, 2014; Samuels and Gouldby, 2009) without double-counting exposure or hazard – includes

no method of shoreline protection other than beach nourishment, and no explicit inclusion of storm recurrence or severity.…"

*Line 29: I think you could really hit this out of the park with some spatial discussion. What are the regional trends? What types of counties seems to be most at risk? A map color coded for risk would also be a great addition.*

We offer that these are the color-coded patterns shown in Figs. 1 & 2. We have opted not to include a spatially explicit map in part because the trajectories of risk become more difficult to render, as opposed to in the matrix format we use. (Secondarily, the matrices also help reinforce that these results, though interesting and robust, are exploratory.)

*Figure 1: Why does it seem that some counties have more rows than others? Are you just labelling some of them? Or do some counties have multiple rows?*

This scaling is explained at the very start of the Results (P7, L18), but we have added this explanation to the caption.

*Figure 5: How are these rates calculated? What is the difference between historical and long-term?*

Explanation of these trajectories (and the relative differences in their calculation) are provided on P3, L28.

*Figures 7 and 9: How are the counts so high in these figures? If you only have 51 counties, shouldn't they be lower? Otherwise you are applying the same county level exposure number across multiple shoreline transects? I would suggest making these figures over counties and not transects. Additionally, is exposure normalized by area? Because otherwise this just shows that bigger counties with more resources for nourishment are adding more property, if I am reading this correctly (not sure if I am reading it correctly because of the count issue I brought up above).*

We have amended the captions to specify that the distributions are transect-level, and have added the following text (in blue) to the beginning of Section 3.2, on "Component relationships" (P8, L26):

Finally, we compared the statistical distributions of exposure in high- and low-hazard counties, and in high- and low-intensity nourishing counties (as an aspect of vulnerability), to examine whether the three components of risk, as we represent them, reflect temporal interrelationships. In keeping with the scaled stripes in Figures 1, 2, and 4, we present these distributions (Figs. 7, 8) at the transect scale rather than the county scale to better represent the contributions of counties by their coastal extents. For example, Queens County, NY, hosts a high density of exposure per alongshore kilometere – very high exposure and a short coastline – and contributes only four transects to the total (Fig. 2). Likewise, because of its size, Dare County, NC, has both high exposure and a longer shoreline, resulting in a lower value of exposure per alongshore kilometre that accounts for over 100 transects of the domain. Overall, Dare County is less densely developed than Queens County. However, our treatment of exposure does overlook concentrated areas of high-density development within otherwise low-density counties – hotspots at which hazard, exposure, and vulnerability (i.e. nourishment activity) may be closely related.

*Figure 8 and 10: The p-value part of the figures is a bit unnecessary. I think you could use words in the results section to describe this. Also the y-axes, particularly in Figure 8, are misleading. It makes it look like the difference in exposure is huge, when in reality it is less than 0.1. Please make axes consistent (across historical and recent) and bigger range on the figure to better represent what is actually happening.*

We have amended these figures for clarity, but also shifted them to the Supplement, given their supportive roles for Figs 7 & 8 (formerly Fig. 9), respectively.

---

## Author Comment (AC2) · 16 Sep 2019

**RESPONSE TO REVIEWERS** – *nhess-2019-159*

13 September 2019

Dear Editors –

We thank you and the two reviewers for your time, and for the opportunity to revise this submission. Here we offer detailed responses to the comments made by both reviewers (whose remarks are *italicised*).

Sincerely –

*Eli Lazarus* (E.D.Lazarus@soton.ac.uk)
*Scott Armstrong*
* * *
**REVIEWER #2 (JLT) Comments**

*Page 4-Line31: . . . that tracks the vulnerability associated with beach width (Vbw) and beach nourishment (Vbn)...*

Corrected.

*Page 5-Lines 5-10: Equation (3) suggests that Vbw is equal to 1 when x=xo. Is 1 just an arbitrary value? If this is the case, I suggest the authors clarify this in the text. Additionally, the authors normalize Vbw by the min and max of Vbw (as we can see in Figure 3 and 6, for instance). Being this the case, would it be easier to write the normalized expression as Vbw = 1-x/xo?*

We have corrected Eq. (3), and clarified that $V_{bw}$ is a normalised value.

*Page 5-Line 7: . . . in 1970 all counties had the same beach width (x). . .. The use of "x" in this case might be misleading. I believe "x" is the beach width at any point in time, not just in 1970.*

Corrected (by deleting reference to $x$).

*Line 15-20: I suggest the authors include the equation used to calculate Vbn. Including this expression will also help to better understand lines 20-32 in the results section (page 7).*

We have added a new Eq. 4 (P5, L25) to show the expression we describe.

*Additionally, I suggest the authors better explaining why as beach nourishment volume and frequency increases, the vulnerability of a coastal community increases. I can see why this is the case, but it might not be intuitive. Is it perhaps due to the community becoming dependent on such practices, which in turn depend on the availability of a limited resource?*

Addressed with new text and citations at P7, L1.

The new text (in blue) reads:

"Like a ratchet, the cumulative beach-nourishment factor ($V_{bn}$) increases each time a county nourishes. This assumption represents the fact that nourishment projects for

shoreline protection (as opposed to reactionary projects for emergency storm response) are cyclical within multi-decadal programmes (NRC 1995, 2014). Nourishment at a given site rarely occurs only once. A community that initiates a nourishment programme will likely depend on periodic nourishment into the future. By comparison, the beach-width factor ($V_{bw}$) is more dynamic, reflecting the oscillatory behaviour of a nourishment cycle at multi-annual time scales by dropping to a minimum after a nourishment project (as the wide beach buffers property from hazard) and then increasing as the nourished beach erodes and coastal properties become more susceptible to hazard."

*Page 7 – Line 15: Would it be useful to mention here that the shoreline erosion rate predicted by bathtub models often underestimates the natural rate of erosion? This is particularly the case in barrier island environments, which are quite common in the region of study included in this manuscript.*

We have added this caveat and an appropriate reference at P4, L15, and at P7, L31:

Our estimation is effectively a "bathtub model" of change, controlled only by topography with no incorporation of wave-driven sediment transport or other shoreline dynamics. Bathtub models tend to underpredict shoreline erosion rates in wave-dominated, sandy barrier settings, such as those of the US Mid-Atlantic (Lorenzo-Trueba and Ashton, 2014; Wolinsky and Murray, 2009).

The alongshore mean rate derived from sea-level rise shows close agreement with the mean "recent" shoreline-change rate, suggesting that our simplified "bathtub" representation of hazard is a reasonable proxy on a multi-decadal time scale (Fig. 5), even though bathtub models tend to underestimate shoreline erosion rates along barrier coastlines (Lorenzo-Trueba and Ashton, 2014; Wolinsky and Murray, 2009).

*Page 7 – Line 18: . . .we ranked each county by its risk. . .*

Corrected.